# Decoupled Contrastive Learning

## Abstract

Contrastive learning (CL) is one of the most successful paradigms for self-supervised learning (SSL). Specifically, contrastive learning treats two augmented "views" of the same sample as positive, pulling them close and treating all other samples as negative to push them far apart. Despite the evident success of CL SSL methods, there are several challenges in the existing methods as they may require special structures, large batches, or huge training epochs, etc. Our motivation in this work is to provide a simple, efficient, and yet competitive contrastive learning baseline. Through both theoretical and empirical studies, we identified a strong negative-positive-coupling (NPC) effect in the widely used cross-entropy loss in CL SSL methods. We hypothesize that the NPC effect may be a major cause of the inefficiency in many contrastive learning methods. By removing the NPC effect, we reach a decoupled contrastive learning (DCL) objective function, which significantly improves the training efficiency. DCL can achieve competitive performance, requiring neither large batches in SimCLR, momentum encoding in Moco, or large epochs. We demonstrate the benefit of DCL in various benchmarks. Further, DCL is also much less sensitive to suboptimal hyperparameters. Notably, our approach achieves 66.9% ImageNet top-1 accuracy with 256 batch size within 200 epochs pre-training, which outperforms its baseline SimCLR by 5.1%. We believe DCL may provide a strong baseline for future contrastive learning-based SSL studies.

## 1 Introduction

As a fundamental task in machine learning, representation learning aims to extract features to reconstruct the raw data fully. It has been regarded as a long-acting goal over the past decades. Recent progress on representation learning has achieved a significant milestone over self-supervised learning (SSL), facilitating feature learning with its competence in exploiting massive raw data without any annotated supervision. In the early stage of SSL, representation learning has focused on exploiting pretext tasks, which are addressed by generating pseudo-labels to the unlabeled data through different transformations, such as solving jigsaw puzzles [1], colorization [2] and rotation prediction [3]. Though these approaches achieve some success in computer vision, there is a large gap between these methods and supervised learning. Recently, there has been a significant advancement in using contrastive learning [4, 5, 6, 7, 8] for self-supervised pre-training, which significantly closes the gap between the SSL method and supervised learning. Contrastive SSL methods, e.g., SimCLR [8], in general, try to pull different views of the same instance close and push different instances far apart in the representation space.

Despite the evident progress of the state-of-the-art contrastive SSL methods, there have been several challenges in future developing this direction: 1) The SOTA models [7] may require unique structures like the momentum encoder and large memory queues, which may complicate the understanding. 2) The contrastive SSL models [8] may depend on large batch size and huge epoch numbers to achieve competitive performance, posing a computational challenge for academia to explore this direction.

Submitted to 35th Conference on Neural Information Processing Systems (NeurIPS 2021). Do not distribute.

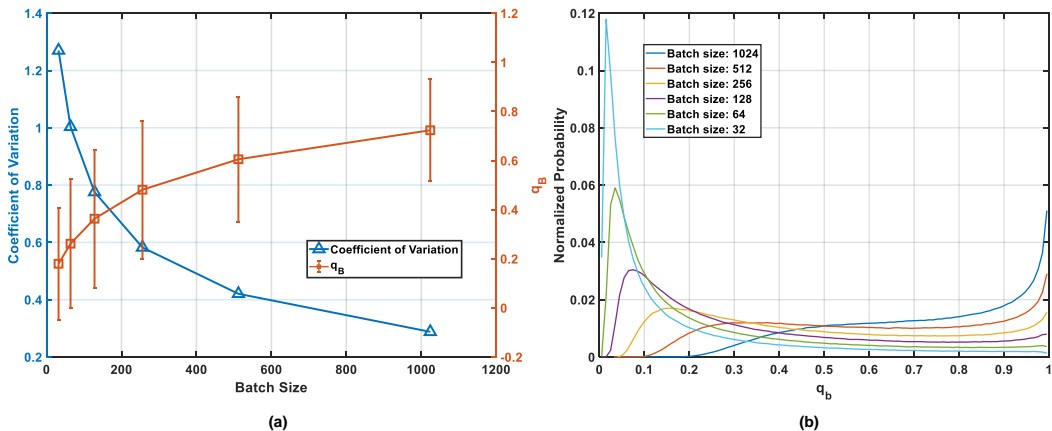

Figure 1: An overview of the batch size issue in the general contrastive approaches: (a) shows the NPC multiplier $q_B$ in different batch sizes. As the large batch size increasing the $q_B$ will approach 1 with a small coefficient of variation. (b) illustrates the distribution of $q_B$.

3) They may be sensitive to hyperparameters and optimizers, introducing additional difficulty to reproduce the results on various benchmarks.

Our motivation in this work is to provide a simple, efficient, and yet competitive contrastive learning baseline. We choose SimCLR as our starting point, given its conceptual simplicity. By analyzing the objective function, we identified a Negative-Positive-Coupling (NPC) multiplier $q_B$ in the gradient as shown in Proposition 1. The NPC multipliers modulate the gradient of each sample, and it mistakenly increases the impact of both negative samples and positive samples, given either of them is more informative. Such a coupling exacerbates when smaller batch sizes are used. By removing the coupling term, we reach a new formulation, the *decoupled contrastive learning* (DCL). The new objective function significantly improves the training efficiency, requires neither large batches, momentum encoding, or large epochs to achieve competitive performance on various different benchmarks. Specifically, DCL reaches 66.9% ImageNet top-1 (linear probing) accuracy with batch size 256, SGD optimizer within 200 epochs. Even if DCL is trained for 100 epochs, it still reaches 64.6% ImageNet top-1 accuracy with batch size 256.

In short, this work makes the following contributions:

1) We provide both theoretical analysis and empirical evidence to show the negative-positive coupling in the gradient of contrastive learning;

2) We introduce a new, decoupled contrastive learning (DCL) objective, which casts off the coupling phenomenon between positive and negative samples in contrastive learning, and significantly improves the training efficiency; Additionally, the proposed DCL objective is less sensitive the several important hyperparameters;

3) We demonstrate our approach via extensive experiments and analysis on both large and small-scale vision benchmarks, with an optimal configuration for the standard SimCLR baseline to have a competitive performance within contrastive approaches.

## 2 Related work

### 2.1 Self-supervised representation learning

Self-supervised representation learning (SSL) aims to learn a robust embedding space from data without human annotation. Previous arts can be roughly categorized into generative and discriminative. Generative approaches, such as autoencoders and adversarial learning, focus on reconstructing images from latent representations [9, 10]. Conversely, recent discriminative approaches, especially contrastive learning-based approaches, have gained the most ground and achieved state-of-the-art standard large-scale image classification benchmarks with increasingly more compute and data augmentations.

## 2.2 Contrastive learning

Contrastive learning (CL) constructs positive and negative sample pairs to extract information from the data itself. In CL, each anchor image in a batch has only one positive sample to construct a positive sample pair [11, 8, 7]. CPC [5] predicts the future output of sequential data by using current output as prior knowledge, which can improve the feature representing the ability of the model. Instance discrimination [4] proposes a non-parametric cross-entropy loss to optimize the model at the instance level. Inv. spread [12] makes use of data augmentation invariants and the spread-out property of instance to learn features. MoCo [7] proposes a dictionary to maintain a negative sample set, thus increasing the number of negative sample pairs. Different from the aforementioned self-supervised CL approaches, [13] proposes a supervised CL that considers all the same categories as positive pairs to increase the utility of images.

## 2.3 Collapsing issue via batch size and negative sample

In CL, the objective is to maximize the mutual information between the positive pairs. However, to avoid the "*collapsing output*", vast quantities of negative samples are needed so that the learning objectives obtain the maximum similarity and have the minimum similarity with negative samples. For instance, in SimCLR [8], training requires many negative samples, leading to a large batch size (i.e., 4096). Furthermore, to optimize such a huge batch, a specially designed optimizer LARS [14] is used. Similarly, MoCo [7] needs a vast queue (i.e., 65536) to achieve competitive performance. BYOL [15] does not collapse output without using any negative samples by considering all the images are positive and to maximize the similarity of "projection" and "prediction" features. On the other hand, Simsam [16] leverages the Siamese network to introduce inductive biases for modeling invariance. With the small batch size (i.e., 256), Simsam is a rival to BYOL (4096). Unlike both approaches that achieved their success through empirical studies, this paper tackles from a theoretical perspective, proving that an intertwined multiplier $q_B$ of positive and negative is the main issue to contrastive learning.

## 3 Decouple negative and positive samples in contrastive learning

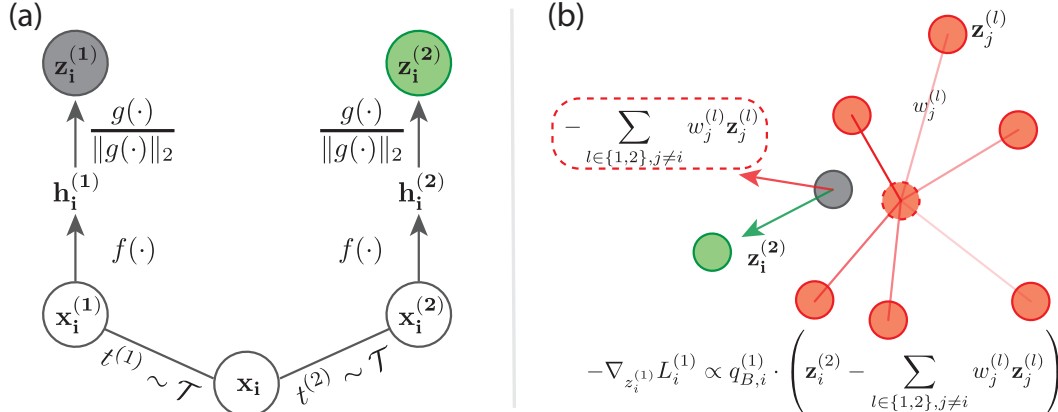

Figure 2: Contrastive learning and negative-positive coupling (NPC). (a) In SimCLR, each sample $\mathbf{x}_i$ has two augmented views $\{\mathbf{x}_i^{(1)}, \mathbf{x}_i^{(2)}\}$. They are encoded by the same encoder $f$ and further projected to $\{\mathbf{z}_i^{(1)}, \mathbf{z}_i^{(2)}\}$ by a normalized MLP. (b) According to Equation 3. For the view $\mathbf{x}_i^{(1)}$, the cross-entropy loss $L_i^{(1)}$ leads to a positive force $\mathbf{z}_i^{(2)}$, which comes from the other view $\mathbf{x}_i^{(2)}$ of $\mathbf{x}$ and a negative force, which is a weighted average of all the negative samples, i.e. $\{\mathbf{z}_j^{(l)} | l \in \{1, 2\}, j \neq i\}$. However, the gradient $-\nabla_{\mathbf{z}_i^{(2)}} L_i^{(1)}$ is proportional to the NPC multiplier.

We choose to start from SimCLR because of its conceptual simplicity. Given a batch of $N$ samples (e.g. images), $\{\mathbf{x}_1, \ldots, \mathbf{x}_N\}$, let $\mathbf{x}_i^{(1)}, \mathbf{x}_i^{(2)}$ be two augmented views of the sample $x_i$ and $B$ be the set of all of the augmented views in the batch, i.e. $B = \{\mathbf{x}_i^{(k)} | k \in \{1, 2\}, i \in [\![1, N]\!]\}$. As

101 shown by Figure 2(a), each of the views $\mathbf{x}_i^{(k)}$ is sent into the same encoder network $f$ and the output

102 $\mathbf{h}_i^{(k)} = f(\mathbf{x}_i^{(k)})$ is then projected by a normalized MLP projector that $\mathbf{z}_i^{(k)} = g(\mathbf{h}_i^{(k)})/\|g(\mathbf{h}_i^{(k)})\|$.

103 For each augmented view $\mathbf{x}_i^{(k)}$, SimCLR solves a classification problem by using the rest of the

104 views in $B$ as targets, and assigns the only positive label to $\mathbf{x}_i^{(u)}$, where $u \neq k$. So SimCLR

105 creates a cross-entropy loss function $L_i^{(k)}$ for each view $\mathbf{x}_i^{(k)}$, and the overall loss function is

106 $L = \sum_{k\in\{1,2\},i\in[\![1,N]\!]} L_i^{(k)}$.

$$L_i^{(k)} = -\log \frac{\exp(\langle \mathbf{z}_i^{(1)}, \mathbf{z}_i^{(2)}\rangle/\tau)}{\exp(\langle \mathbf{z}_i^{(1)}, \mathbf{z}_i^{(2)}\rangle/\tau) + \sum_{l\in\{1,2\},j\in[\![1,N]\!],j\neq i}\exp(\langle \mathbf{z}_i^{(k)}, \mathbf{z}_j^{(l)}\rangle/\tau)} \tag{1}$$

107 *Proposition* 1. There exists a negative-positive coupling (NPC) multiplier $q_{B,i}^{(1)}$ in the gradient of

108 $L_i^{(1)}$:

$$\begin{cases} -\nabla_{\mathbf{z}_i^{(1)}} L_i^{(1)} = \frac{q_{B,i}^{(1)}}{\tau}\left[\mathbf{z}_i^{(2)} - \sum_{l\in\{1,2\},j\in[\![1,N]\!],j\neq i}\frac{\exp\langle \mathbf{z}_i^{(1)},\mathbf{z}_j^{(l)}\rangle/\tau}{\sum_{q\in\{1,2\},j\in[\![1,N]\!],j\neq i}\exp(\langle \mathbf{z}_i^{(1)},\mathbf{z}_j^{(q)}\rangle/\tau)}\cdot \mathbf{z}_j^{(l)}\right] \\ -\nabla_{\mathbf{z}_i^{(2)}} L_i^{(1)} = \frac{q_{B,i}^{(1)}}{\tau}\cdot \mathbf{z}_i^{(1)} \\ -\nabla_{\mathbf{z}_j^{(l)}} L_i^{(1)} = -\frac{q_{B,i}^{(1)}}{\tau}\frac{\exp\langle \mathbf{z}_i^{(1)},\mathbf{z}_j^{(l)}\rangle/\tau}{\sum_{q\in\{1,2\},j\in[\![1,N]\!],j\neq i}\exp(\langle \mathbf{z}_i^{(1)},\mathbf{z}_j^{(q)}\rangle/\tau)}\cdot \mathbf{z}_i^{(1)} \end{cases} \tag{2}$$

109 where the NPC multiplier $q_{B,i}^{(1)}$ is:

$$q_{B,i}^{(1)} = 1 - \frac{\exp(\langle \mathbf{z}_i^{(1)}, \mathbf{z}_i^{(2)}\rangle/\tau)}{\sum_{q\in\{1,2\},j\in[\![1,N]\!],j\neq i}\exp(\langle \mathbf{z}_i^{(1)}, \mathbf{z}_j^{(q)}\rangle/\tau)} \tag{3}$$

110 Due to the symmetry, a similar NPC multiplier $q_{B,i}^{(k)}$ exists in the gradient of $L_i^{(k)}, k \in \{1,2\}, i \in$

111 $[\![1,N]\!]$.

112 As we can see, all of the partial gradients in Equation 2 are modified by the common NPC multiplier

113 $q_{B,i}^{(k)}$ in Equation 3. Equation 3 makes intuitive sense: 1) When a positive sample pair $\{\mathbf{z}_i^{(1)}, \mathbf{z}_i^{(2)}\}$ are

114 farther, the corresponding NPC multiplier $q_{B,i}^{(1)}$ is larger. This will makes the overall gradient larger.

115 Otherwise, the gradient is smaller. 2) When a negative sample is closer to $\mathbf{z}_i^{(1)}$, it makes $q_{B,i}^{(1)}$ larger.

116 Overall, the intuition here is that a positive sample farther from the target or a negative sample closer

117 to the target is more informative. However, the positive samples and negative samples are strongly

118 coupled. An outlier positive sample also makes the gradient from the negative samples significantly

119 larger and vice versa.

120 Figure 1(b) shows the NPC multiplier $q_B$ distribution shift w.r.t. different batch sizes for a pre-trained

121 SimCLR baseline model. While all of the shown distributions have prominent fluctuation, the smaller

122 batch size makes $q_B$ cluster towards 0, while the larger batch size pushes the distribution towards

123 $\delta(1)$. Figure 1(a) shows the averaged NPC multiplier $\langle q_B \rangle$ changes w.r.t. the batch size and the

124 relative fluctuation. The small batch sizes introduce significant NPC fluctuation. Based on this

125 observation, we propose to remove the NPC multipliers from the gradients, which corresponds to the

126 case $q_{B,N\to\infty}$. This leads to the decoupled contrastive learning formulation.

127 *Proposition* 2. Removing the positive pair from the denominator of Equation 2 leads to a decoupled

128 contrastive learning loss. If we remove the NPC multiplier $q_{B,i}^{(k)}$ from Equation 2, we reach a

129 decoupled contrastive learning loss $L_{DC} = \sum_{k\in\{1,2\},i\in[\![1,N]\!]} L_{DC,i}^{(k)}$, where $L_{DC,i}^{(k)}$ is:

$$L_{DC,i}^{(k)} = -\log \frac{\exp(\langle \mathbf{z}_i^{(1)}, \mathbf{z}_i^{(2)}\rangle/\tau)}{\cancel{\exp(\langle \mathbf{z}_i^{(1)}, \mathbf{z}_i^{(2)}\rangle/\tau)} + \sum_{l\in\{1,2\},j\in[\![1,N]\!],j\neq i}\exp(\langle \mathbf{z}_i^{(k)}, \mathbf{z}_j^{(l)}\rangle/\tau)} \tag{4}$$

$$= -\langle \mathbf{z}_i^{(1)}, \mathbf{z}_i^{(2)}\rangle/\tau + \log \sum_{l\in\{1,2\},j\in[\![1,N]\!],j\neq i}\exp(\langle \mathbf{z}_i^{(k)}, \mathbf{z}_j^{(l)}\rangle/\tau) \tag{5}$$

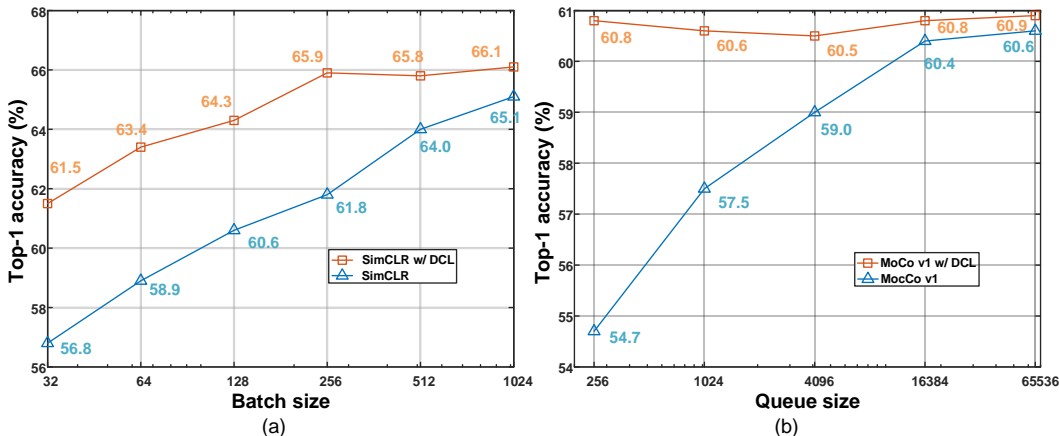

Figure 3: Comparisons on ImageNet-1K with/without DCL under different numbers of (a): batch sizes for SimCLR [8] and (b): queues for MoCo [7]. Without DCL, the top-1 accuracy significantly drops when batch size (SimCLR) or queues (MoCo) becomes very small.

The proofs of Proposition 1 and 2 are given in Appendix. Further, we can generalize the loss function $L_{DC}$ to $L_{DCW}$ by introducing a weighting function for the positive pairs i.e. $L_{DCW} = \sum_{k \in \{1,2\}, i \in [\![1,N]\!]} L_{DCW,i}^{(k)}$.

$$L_{DCW,i}^{(k)} = -w(\mathbf{z}_i^{(1)}, \mathbf{z}_i^{(2)})(\langle \mathbf{z}_i^{(1)}, \mathbf{z}_i^{(2)} \rangle / \tau) + \log \sum_{l \in \{1,2\}, j \in [\![1,N]\!], j \neq i} \exp(\langle \mathbf{z}_i^{(k)}, \mathbf{z}_j^{(l)} \rangle / \tau) \quad (6)$$

where we can intuitively choose $w$ to be a negative von Mises-Fisher weighting function that $w(\mathbf{z}_i^{(1)}, \mathbf{z}_i^{(2)}) = 2 - \frac{\exp(\langle \mathbf{z}_i^{(1)}, \mathbf{z}_i^{(2)} \rangle / \sigma)}{\mathrm{E}_i \left[ \exp(\langle \mathbf{z}_i^{(1)}, \mathbf{z}_i^{(2)} \rangle / \sigma) \right]}$ and $\mathrm{E}[w] = 1$. $L_{DC}$ is a special case of $L_{DCW}$ and we can see that $\lim_{\sigma \to \infty} L_{DCW} = L_{DC}$. The intuition behind $w(\mathbf{z}_i^{(1)}, \mathbf{z}_i^{(2)})$ is that there is more learning signal when a positive pair of samples are far from each other.

# 4 Experiments

This section evaluates our proposed decoupled contrastive learning (DCL) empirically and compares it to the general contrastive learning methods. We summarize our experiments and analysis as the following: (1) our proposed work significantly outperforms the general contrastive learning on large and small-scale vision benchmarks; (2) we show the better version of DCL: LDCW could further improve the representation quality. (3) we further analyze our DCL with few learning epochs, which shows fast convergence of the proposed DCL. Detailed experimental settings can be found in the Appendix.

## 4.1 Implementation details

To understand the effect of the sample decoupling, we consider our proposed DCL, which is based on the general contrastive learning, where model optimization is irrelevant to the size of batches (i.e., negative samples). Extensive experiments and analysis are demonstrated on large-scale benchmarks: ImageNet-1K [19], ImageNet-100 [6], and small-scale benchmark: CIFAR [20], and STL10 [21]. Note that all of our experiments are conducted with 8 Nvidia V100 GPUs on a single machine.

**ImageNet**  For a fair comparison on ImageNet data, we implement our proposed decoupled structure, DCL by following SimCLR [8] with ResNet-50 [22] as the encoder backbone and use cosine annealing schedule. We set the temperature $\tau$ to 0.1 and the latent vector dimension to 128. Following [23], we evaluate the pre-trained models by training a linear classifier with frozen learned embedding on ImageNet data. We further consider evaluating our approach on ImageNet-100, a selected subset of 100 classes of ImageNet-1K.

Table 1: Comparisons with/without DCL under different numbers of batch sizes from 32 to 512. Results show the effectness of DCL on four widely used benchmarks. The performance of DCL keeps steadier than the SimCLR baseline while the batch size is varied.

| Dataset | ImageNet-100 (linear) | | | | | CIFAR10 (kNN) | | | | |
|---|---|---|---|---|---|---|---|---|---|---|
| Batch Size | 32 | 64 | 128 | 256 | 512 | 32 | 64 | 128 | 256 | 512 |
| SimCLR [8] | 74.2 | 77.6 | 79.3 | 80.7 | 81.3 | 78.9 | 80.4 | 81.1 | 81.4 | 81.3 |
| SimCLR w/ DCL | **80.8** | **82.0** | **81.9** | **83.1** | **82.8** | **83.7** | **84.4** | **84.4** | **84.2** | **83.5** |
| Dataset | CIFAR100 (kNN) | | | | | STL10 (kNN) | | | | |
| Batch Size | 32 | 64 | 128 | 256 | 512 | 32 | 64 | 128 | 256 | 512 |
| SimCLR [8] | 49.4 | 50.3 | 51.8 | 52 | 52.4 | 74.1 | 76.2 | 76.9 | 77.3 | 77.6 |
| SimCLR w/ DCL | **51.1** | **54.3** | **54.6** | **54.9** | **55** | **82.0** | **82.8** | **81.8** | **81.2** | **81.0** |

Table 2: kNN top-1 accuracy (%) comparison of SSL approaches on small-scale benchmarks: CIFAR10, CIFAR100, and STL10. Results show that DCL consistently improves its SimCLR baseline. With multi-cropping [17], our DCLW reaches competitive performance within other contrastive learning approaches [8, 7, 4, 12, 18].

| kNN (top-1) | SimCLR | MoCo | MoCo + CLD | NPID | NPID + CLD | Inv. Spread | Exemplar | DCL | DCLW w/ mcrop |
|---|---|---|---|---|---|---|---|---|---|
| CIFAR10 | 81.4 | 82.1 | 87.5 | 80.8 | 86.7 | 83.6 | 76.5 | 84.1 | **87.8** |
| CIFAR100 | 52.0 | 53.1 | 58.1 | 51.6 | 57.5 | N/A | N/A | 54.9 | **58.8** |
| STL10 | 77.3 | 80.8 | 84.3 | 79.1 | 83.6 | 81.6 | 79.3 | 81.2 | **84.1** |

**CIFAR and STL10** For CIFAR10, CIFAR100, and STL10, ResNet-18 [22] is used as the encoder architecture. We set the temperature $\tau$ to 0.07. All models are trained for 200 epochs with SGD optimizer and a base $lr = 0.03 * batchsize/256$. We follow NPID [4] on using $k = 200$ nearest neighbor (kNN) classifier. Note that on STL10, we follow [24] to use both $train$ set and $unlabeled$ set for model pre-training.

## 4.2 Experiments and analysis

**DCL on ImageNet** This section illustrates the effect of our DCL under different batch sizes and queues. The initial setup is to have 1024 batch size (SimCLR [8]) and 65536 queues (MoCo [7]) and gradually reduce the batch size (SimCLR) and queue (MoCo) to show the corresponding top-1 accuracy by linear evaluation. Figure 3 indicates that without DCL, the top-1 accuracy drastically drops when batch size (SimCLR) or queue (MoCo) becomes very small. While with DCL, the performance keeps steadier than baselines (SimCLR: $-4.1\%$ vs. $-8.3\%$, MoCo: $-0.4\%$ vs. $-5.9\%$).

Specifically, Figure 3 further shows that in SimCLR, the performance with DCL improves from $61.8\%$ to $65.9\%$ under 256 batch size; MoCo with DCL improves from $54.7\%$ to $60.8\%$ under 256 queues. The comparison fully demonstrates the necessity of DCL, especially when the number of negatives is small. Although batch size is increased to 1024, we also note that our DCL ($66.1\%$) still improves over the SimCLR baseline ($65.1\%$).

We further observe the same phenomenon on ImageNet-100 data. Table 1 shows that, while with DCL, the performance only drops $2.3\%$ compare to the SimCLR baseline of $7.1\%$.

In summary, it is worth noting that, while the batch size is small, the strength of $q_{B,i}$, which is used to push the negative samples away from the positive sample, is also relatively weak. This phenomenon tends to reduce the efficiency of learning representation. While taking advantage of DCL alleviates the performance gap between small and large batch sizes. Hence, through the analysis, we find out DCL can simply tackle the batch size issue in contrastive learning. With this considerable advantage given by DCL, general SSL approaches can be implemented with fewer computational resources or lower standard platforms.

Table 3: Comparisons between SimCLR baseline, DCL, and DCLW. Results indicate that DCL improves the performance of baseline, and DCLW further provides an extra boost. Note that results are under the batch size 256 and epoch 200. All of models are both trained and evaluated with same experimental settings.

| | Baseline | DCL | DCLW |
|---|---|---|---|
| CIFAR10 | 81.8 | 84.2 (**+3.1**) | **84.8 (+3.7)** |
| CIFAR100 | 51.8 | 54.6 (**+2.8**) | **54.9 (+3.1)** |
| ImageNet-100 | 79.3 | 81.9 (**+2.6**) | **82.8 (+3.5)** |
| ImageNet-1K | 61.8 | 65.9 (**+4.1**) | **66.9 (+5.1)** |

Table 4: ImageNet-1K top-1 accuracy (%) on SimCLR and MoCo v2 with/without DCL under few training epochs. We further list results under 200 epochs for clear comparison. With DCL, the performance of SimCLR trained under 100 epochs nearly reaches its performance under 200 epochs. The MoCo v2 with DCL also reaches higher accuracy than the baseline under 100 epochs.

| | SimCLR[8] | SimCLR w/ DCL | MoCo v2[25] | MoCo v2 w/ DCL |
|---|---|---|---|---|
| 100 epoch | 57.5 | 64.6 | 63.6 | 64.4 |
| 200 epoch | 61.8 | 65.9 | 67.5 | 67.7 |

**DCL on CIFAR and STL10**   In Table 1 and Table 3, it is observed that DCL also demonstrates its effectiveness on small-scale benchmarks. In summary, DCL outperforms its baseline by 3.1% (CIFAR10) and 2.8% (CIFAR100) and keeps the performance relatively steady under batch size 256. We also improve the kNN accuracy of the SimCLR baseline on STL10 by 3.9%.

**Decoupled objective with re-weighting DCLW**   We only replace $L_{DC}$ with $L_{DCW}$ with no possible advantage from additional tricks. That is, both our approach and the baselines apply the same training instruction of the OpenSelfSup benchmark [23] for fairness. Note that we empirically choose $\sigma = 0.5$ in the experiments.

Results in Table 3 indicates that, DCLW achieves extra 5.1% (ImageNet-1K), 3.5% (ImageNet-100) gains compared to the baseline. For CIFAR data, extra 3.7% (CIFAR10), 3.1% are gained from the addition of DCLW. It is worth to note that, trained with 200 epochs, our DCLW reaches 66.9% with batch size 256, surpassing the SimCLR [8] baseline: 66.2% with batch size 8192.

## 4.3   Small-scale benchmark results: STL10, CIFAR10, and CIFAR100

For STL10, CIFAR10, and CIFAR100, we implement our DCL with ResNet-18 [22] as encoder backbone by following small-scale benchmark of CLD [24]. All the models are trained for 200 epochs with 256 batch size and evaluate by using kNN accuracies ($k = 200$).

Results in Table 2 indicates that, our DCLW with multi-cropping [17] consistently outperforms the state-of-the-art baselines on CIFAR10, STL10, and CIFAR100. Our DCL also demonstrates its capability while comparing against other baselines. More analysis of large-scale benchmarks can be found in Appendix.

## 4.4   Ablations

We perform extensive ablations on the hyperparameters of our DCL and DCLW on both ImageNet data and other small-scale data, i.e., CIFAR10, CIFAR100, and STL10. By seeking better configurations empirically, we see that our approach gives consistent gains over the standard SimCLR baseline. In other ablations, we see that our DCL achieves more gains over both SimCLR and MoCo v2, i.e., contrastive learning baselines, also when training for 100 epochs only.

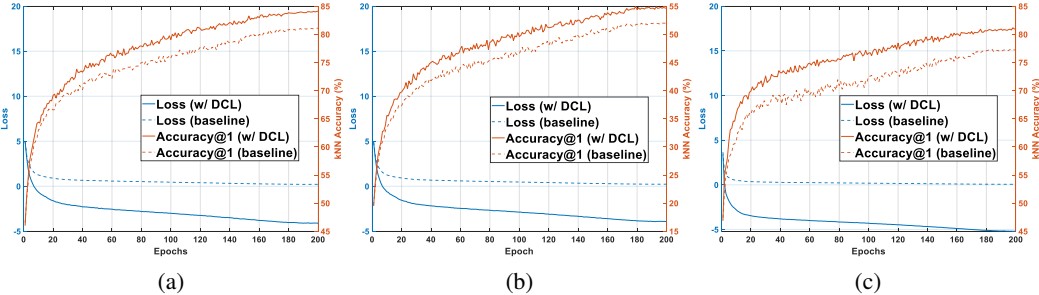

(a)          (b)          (c)

Figure 4: During the SSL pre-training, DCL speeds up the model convergence and provides better performance than the baseline on CIFAR and STL10 data.

**Few learning epochs** Our DCL is inspired by the traditional contrastive learning framework, which needs a large batch size, long learning epochs to achieve higher performance. The previous state-of-the-art, SimCLR [8], heavily rely on large quantities of learning epochs to obtain high top-1 accuracy. (e.g., 69.3% with up to 1000 epochs). The purpose of our DCL is to achieve higher learning efficiency with few learning epochs. We demonstrate the effectiveness of DCL in contrastive learning frameworks SimCLR and MoCo v2. We choose the batch size of 256 (queue of 65536) as the baseline and train the model with only 100 epochs instead of the normal number of 200. We make sure other parameter settings are the same for a fair comparison. Table 4 shows the result on ImageNet-1K using linear evaluation. With DCL, SimCLR can achieve 64.6% top-1 accuracy with only 100 epochs compared to SimCLR baseline: 57.5%; MoCo v2 with DCL reaches 64.4% compared to MoCo v2 baseline: 63.6% with 100 epochs pre-training.

We further demonstrate that, with DCL, learning representation becomes faster during the early stage of training. The reason is that DCL successfully solves the decoupled issue between positive and negative pairs. Figure 4 (a), (b), and (c), show that our DCL improves the speed of convergence and reaches higher performance than the baseline on CIFAR and STL10 data.

## 5 Conclusion

In this paper, we identify the negative-positive-coupling (NPC) effect in SimCLR. By removing the NPC effect, we reach a new objective function, decoupled contrastive learning (DCL). The proposed DCL loss function requires minimal modification to the SimCLR baseline and provides efficient, reliable, and nontrivial performance improvement on various benchmarks. Given the conceptual simplicity of DCL and that it requires neither momentum encoding, large batch sizes, or long epochs to reach competitive performance, we wish that DCL can serve as a strong baseline for the contrastive-based SSL methods.

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
