# A Appendix

Optionally include extra information (complete proofs, additional experiments and plots) in the

appendix. This section will often be part of the supplemental material.

## A.1 Proof of proposition 1

**Proposition 1.** There exists a negative-positive coupling (NPC) multiplier $q_{B,i}^{(1)}$ in the gradient of

$L_i^{(1)}$:

$$
\begin{cases}
-\nabla_{\mathbf{z}_i^{(1)}} L_i^{(1)} = \frac{q_{B,i}^{(1)}}{\tau} \left[ \mathbf{z}_i^{(2)} - \sum_{l \in \{1,2\}, j \in [\![1,N]\!], j \neq i} \frac{\exp \langle \mathbf{z}_i^{(1)}, \mathbf{z}_j^{(l)} \rangle / \tau}{\sum_{q \in \{1,2\}, j \in [\![1,N]\!], j \neq i} \exp(\langle \mathbf{z}_i^{(1)}, \mathbf{z}_j^{(q)} \rangle / \tau)} \cdot \mathbf{z}_j^{(l)} \right] \\
-\nabla_{\mathbf{z}_i^{(2)}} L_i^{(1)} = \frac{q_{B,i}^{(1)}}{\tau} \cdot \mathbf{z}_i^{(1)} \\
-\nabla_{\mathbf{z}_j^{(l)}} L_i^{(1)} = -\frac{q_{B,i}^{(1)}}{\tau} \frac{\exp \langle \mathbf{z}_i^{(1)}, \mathbf{z}_j^{(l)} \rangle / \tau}{\sum_{q \in \{1,2\}, j \in [\![1,N]\!], j \neq i} \exp(\langle \mathbf{z}_i^{(1)}, \mathbf{z}_j^{(q)} \rangle / \tau)} \cdot \mathbf{z}_i^{(1)}
\end{cases}
$$

where the NPC multiplier $q_{B,i}^{(1)}$ is:

$$
q_{B,i}^{(1)} = 1 - \frac{\exp(\langle \mathbf{z}_i^{(1)}, \mathbf{z}_i^{(2)} \rangle / \tau)}{\sum_{q \in \{1,2\}, j \in [\![1,N]\!], j \neq i} \exp(\langle \mathbf{z}_i^{(1)}, \mathbf{z}_j^{(q)} \rangle / \tau)}
$$

Due to the symmetry, a similar NPC multiplier $q_{B,i}^{(k)}$ exists in the gradient of $L_i^{(k)}, k \in \{1,2\}, i \in$

$[\![1,N]\!]$.

*Proof.*

$$
-\nabla_{\mathbf{z}_i^{(1)}} L_i^{(1)} = \frac{\mathbf{z}_i}{\tau} - \frac{1}{Y} \cdot \exp(\langle \mathbf{z}_i^{(1)}, \mathbf{z}_i^{(2)} \rangle / \tau) \cdot \frac{\mathbf{z}_i^{(2)}}{\tau} - \frac{1}{Y} \cdot \sum_{q \in \{1,2\}, j \in [\![1,N]\!], j \neq i} \exp(\langle \mathbf{z}_i^{(1)}, \mathbf{z}_j^{(q)} \rangle / \tau) \frac{\mathbf{z}_j^{(q)}}{\tau}
$$

$$
= (1 - \frac{1}{Y} \cdot \exp(\langle \mathbf{z}_i^{(1)}, \mathbf{z}_i^{(2)} \rangle / \tau)) \frac{\mathbf{z}_i^{(2)}}{\tau} - \frac{1}{Y} \cdot \sum_{q \in \{1,2\}, j \in [\![1,N]\!], j \neq i} \exp(\langle \mathbf{z}_i^{(1)}, \mathbf{z}_j^{(q)} \rangle / \tau) \frac{\mathbf{z}_j^{(q)}}{\tau}
$$

$$
= \frac{1}{\tau} (1 - \frac{1}{Y} \cdot \exp(\langle \mathbf{z}_i^{(1)}, \mathbf{z}_i^{(2)} \rangle / \tau)) \left[ \mathbf{z}_i^{(2)} - \sum_{q \in \{1,2\}, j \in [\![1,N]\!], j \neq i} \frac{\exp(\langle \mathbf{z}_i^{(1)}, \mathbf{z}_j^{(q)} \rangle / \tau)}{U} \cdot \mathbf{z}_j^{(q)} \right]
$$

$$
= \frac{q_{B,i}^{(1)}}{\tau} \left[ \mathbf{z}_i^{(2)} - \sum_{q \in \{1,2\}, j \in [\![1,N]\!], j \neq i} \frac{\exp(\langle \mathbf{z}_i^{(1)}, \mathbf{z}_j^{(q)} \rangle / \tau)}{U} \cdot \mathbf{z}_j^{(q)} \right]
$$

where $Y = \exp(\langle \mathbf{z}_i^{(1)}, \mathbf{z}_i^{(2)} \rangle / \tau) + \sum_{q \in \{1,2\}, j \in [\![1,N]\!], j \neq i} \exp(\langle \mathbf{z}_i^{(1)}, \mathbf{z}_j^{(q)} \rangle / \tau)$, $U =$

$\sum_{q \in \{1,2\}, j \in [\![1,N]\!], j \neq i} \exp(\langle \mathbf{z}_i^{(1)}, \mathbf{z}_j^{(q)} \rangle / \tau)$.

$$
-\nabla_{\mathbf{z}_i^{(2)}} L_i^{(1)} = \frac{1}{\tau} \mathbf{z}_i^{(1)} - \frac{1}{Y} \exp(\langle \mathbf{z}_i^{(1)}, \mathbf{z}_i^{(2)} \rangle / \tau) \cdot \frac{\mathbf{z}_i^{(1)}}{\tau}
$$

$$
= \frac{q_{B,i}^{(1)}}{\tau} \cdot \mathbf{z}_i^{(1)}
$$

$$
-\nabla_{\mathbf{z}_j^{(l)}} L_i^{(1)} = \frac{1}{Y} \exp(\langle \mathbf{z}_i^{(1)}, \mathbf{z}_j^{(q)} \rangle / \tau) \cdot \frac{\mathbf{z}_i^{(1)}}{\tau}
$$

$$
= \frac{q_{B,i}^{(1)}}{\tau} \cdot \frac{\exp(\langle \mathbf{z}_i^{(1)}, \mathbf{z}_j^{(q)} \rangle / \tau)}{U} \mathbf{z}_i^{(1)}
$$

$\square$

## A.2 Proof of proposition 2

**Proposition 2.** Removing the positive pair from the denominator of Equation 2 leads to a decoupled contrastive learning loss. If we remove the NPC multiplier $q_{B,i}^{(k)}$ from Equation 2, we reach a decoupled contrastive learning loss $L_{DC} = \sum_{k \in \{1,2\}, i \in [\![1,N]\!]} L_{DC,i}^{(k)}$, where $L_{DC,i}^{(k)}$ is:

$$
L_{DC,i}^{(k)} = -\log \frac{\exp(\langle \mathbf{z}_i^{(1)}, \mathbf{z}_i^{(2)} \rangle / \tau)}{\exp(\langle \mathbf{z}_i^{(1)}, \mathbf{z}_i^{(2)} \rangle / \tau) + \sum_{l \in \{1,2\}, j \in [\![1,N]\!], j \neq i} \exp(\langle \mathbf{z}_i^{(k)}, \mathbf{z}_j^{(l)} \rangle / \tau)}
$$

$$
= -\langle \mathbf{z}_i^{(1)}, \mathbf{z}_i^{(2)} \rangle / \tau + \log \sum_{l \in \{1,2\}, j \in [\![1,N]\!], j \neq i} \exp(\langle \mathbf{z}_i^{(k)}, \mathbf{z}_j^{(l)} \rangle / \tau)
$$

*Proof.* By removing the positive term the denominator of Equation 4, we can repeat the procedure in the proof of Proposition 1 and see that the coupling term disappears.

□

## A.3 Linear classification on ImageNet-1K

Top-1 accuracies of linear evaluation in Table 5 shows that, we compare with the state-of-the-art SSL approaches on ImageNet-1K. For fairness, we list the batch size and learning epoch of each individual approach, which are shown in the original paper. During pre-training, our DCL is based on a ResNet-50 backbone, with two views with size $224 \times 224$. Without relatively huge batch sizes or other pre-training schemes, i.e., momentum encoder, clustering, and prediction head, our DCL relies on its simplicity to reach competitive performance. We report both 200-epoch and 400-epoch versions of our DCL. It achieves 69.5% under the batch size of 256 and 400-epoch pre-training, which is better than SimCLR [8] in their optimal case, i.e., batch size of 4096, and 1000-epoch. Note that SwAV [26], BYOL [15], SimCLR [8], and PIRL [27] need huge batch size of 4096, and SwAV [17] further applies multi-cropping as generating extra views to reach optimal performance.

Table 5: ImageNet-1K top-1 accuracies (%) of linear classifiers trained on representations of different SSL methods.

| Method | Architecture | Param. (M) | Batch size | Epochs | Top-1 (%) |
|---|---|---|---|---|---|
| Relative-Loc. [28] | ResNet-50 | 24 | 256 | 200 | 49.3 |
| Rotation-Pred. [3] | ResNet-50 | 24 | 256 | 200 | 55.0 |
| DeepCluster [26] | ResNet-50 | 24 | 256 | 200 | 57.7 |
| NPID [4] | ResNet-50 | 24 | 256 | 200 | 56.5 |
| Local Agg. [29] | ResNet-50 | 24 | 256 | 200 | 58.8 |
| MoCo [7] | ResNet-50 | 24 | 256 | 200 | 60.6 |
| SimCLR [8] | ResNet-50 | 28 | 256 | 200 | 61.8 |
| CMC [6] | ResNet-50$_{L+ab}$ | 47 | 256 | 280 | 64.1 |
| MoCo v2 [25] | ResNet-50 | 28 | 256 | 200 | 67.5 |
| SwAV [17] | ResNet-50 | 28 | 4096 | 200 | 69.1 |
| SimSiam [16] | ResNet-50 | 28 | 256 | 200 | 70.0 |
| InfoMin [30] | ResNet-50 | 28 | 256 | 200 | 70.1 |
| BYOL [15] | ResNet-50 | 28 | 4096 | 200 | 70.6 |
| DCL | ResNet-50 | 28 | 256 | 200 | 67.8 |
| PIRL [27] | ResNet-50 | 24 | 256 | 800 | 63.6 |
| SimCLR [8] | ResNet-50 | 28 | 4096 | 1000 | 69.3 |
| MoCo v2 [25] | ResNet-50 | 28 | 256 | 800 | 71.1 |
| SwAV [17] | ResNet-50 | 28 | 4096 | 400 | 70.7 |
| SimSiam [16] | ResNet-50 | 28 | 256 | 800 | 71.3 |
| DCL | ResNet-50 | 28 | 256 | 400 | 69.5 |

### A.4 Implementation details

**DCL augmentations.** We follow the settings of SimCLR [8] to set up the data augmentations. We use $RandomResizedCrop$ with scale in [0.08, 1.0] and follow by $RandomHorizontalFlip$. Then, $ColorJittering$ with strength in [0.8, 0.8, 0.8, 0.2] with probability of 0.8, and $RandomGrayscale$ with probability of 0.2. $GaussianBlur$ includes Gaussian kernel with standard deviation in [0.1, 2.0].

**Linear evaluation.** Following the open-sourced project, OpenSelfSup [23], we first train the linear classifier with batch size 256 for 100 epochs. We use the SGD optimizer with momentum = 0.9, and weight decay = 0. The base $lr$ is set to 30.0 and decay by 0.1 at epoch [60, 80]. We further demonstrate the linear evaluation protocol of SimSiam [16], which raises the batch size to 4096 for 90 epochs. The optimizer is switched to LARS optimizer with base $lr = 1.2$ and cosine decay schedule. The momentum and weight decay are remained unchanged. We found the second one slightly improves the performance.