# OpenReview forum: "Decoupled Contrastive Learning"
_NeurIPS.cc/2021/Conference — NeurIPS 2021 Submitted_

### Official Review · Reviewer_mww5 · 2021-07-13

**Rating:** 6
**Confidence:** 3

**Summary:**

This paper gives a theoretical analysis of the contrastive loss to identify a "negative-positive coupling multiplier" term. By removing this term from the loss function, the authors achieved better performance over SimCLR and MOCO, particularly when using small batch sizes.

**Limitations And Societal Impact:**

This is not applicable to this paper.

**Main Review:**

**Pros:**

1. The authors identified a common multiplier term in the gradients of the well known contrastive loss, and observed that this multiplier (NPC) behaved differently for small vs large batch sizes. Since NPC fluctuates wildly for small batches, but converges to 1 for large batches, they propose to set NPC to 1 during training. The technique is very simple and does not require further tricks.

2. There is also an improved version that uses a weighting between positive and negative terms in the loss function. By choosing a weighting function cleverly they avoid having more hyperparameters to tune. (The temperature parameter needed to be re-tuned though.)

3. For small batch sizes, the performance and training efficiency on various datasets exceed SimCLR baselines. This is nice because these are two known disadvantages of contrastive learning (bad at small batches, slow training).

4. Writing is quite clear.

**Cons:**

I have some concerns with regards to the significance of the results. Contrastive learning is known to not do well at small batch sizes. Follow-up methods such as BYOL have achieved much better results at small batch sizes. While it is very nice that this paper has improved over SimCLR and MOCO at small batches, it still does not match the later SSL methods, and practical applications could be limited. The benefits may also disappear when models and training are scaled up.

**Nit:**

Line 127 - “Removing the positive pair from the denominator of Equation 2” - I think this should be Equation 1?


**Time Spent Reviewing:**

2

---

> ### Author Response · Authors · 2021-08-10
> **Response to Reviewer mww5**
>
> #### __Q4-1:__
> Contrastive learning is known to not do well at small batch sizes. Follow-up methods such as BYOL have achieved much better results at small batch sizes. While it is very nice that this paper has improved over SimCLR and MOCO at small batches, it still does not match the later SSL methods, and practical applications could be limited.
>
> #### __A4-1:__
> This is an important point. On the one hand, while the non-contrastive methods have achieved better performance on large-scale benchmarks like ImageNet, competitive contrastive methods, like NNCLR [1], have recently been proposed. The DCL method can be potentially combined with the method NNCLR to achieve further improvement. The SOTA SSL speech models, e.g., wav2vec V2 [2], still uses contrastive loss in the objective; DCL can also potentially improve these large-scale benchmarks.
>     On the other hand, to the best of our knowledge, there is not a consensus that non-contrastive methods would lead to the SOTA universally on different datasets. In fact, in the following, we can use CIFAR-10 as an example to show that DCL achieves competitive results compared to BYOL, SimSiam, and Barlow Twins. Last but not least, we acknowledge that generalizing the insights from DCL to the non-contrastive methods is an exciting direction.
>
> |   CIFAR-10 (ResNet-18)  	| Batch Size 	| Epoch 	| kNN Accuracy 	|
> |:-----------:	|:----------:	|:-----:	|:------------:	|
> |     BYOL*    	|     128    	|  200  	|     __85\%__     	|
> |   SimSiam*   	|     128    	|  200  	|     73\%     	|
> | BarlowTwins* 	|     128    	|  200  	|     84\%     	|
> |     DCL     	|     128    	|  200  	|     84\%    	|
> |
> |     BYOL*    	|     512    	|  200  	|     __84\%__     	|
> |   SimSiam*   	|     512    	|  200  	|     81\%     	|
> | BarlowTwins* 	|     512    	|  200  	|     78\%     	|
> |     DCL     	|     512    	|  200  	|     __84\%__     	|
>
> \* The method is implemented by [3].
>
>
> #### __Q4-2:__
> The benefits may also disappear when models and training are scaled up.
>
> #### __A4-2:__
> We acknowledge this limit. We think the proposed DCL still provides important value: 1) The analysis and empirical results can provide insights into the contrastive learning methods. This may contribute to further modeling advancement. 2) Contrastive SSL methods are notorious for their resource consumption, which actually poses a roadblock for many academic environments on studying the contrastive methods. We hope that the DCL can provide a competitive method with less requirement. 3) In resource-constrained settings, e.g., SSL on edge devices or federated contrastive learning, the proposed DCL may also provide important improvement.
>
> #### __Nit:__ Line 127 - “Removing the positive pair from the denominator of Equation 2” - I think this should be Equation 1?
> Thank you for pointing this out! We have fixed it in the revision.
>
>
> #### __Q4-3:__
> Limitations And Societal Impact
>
> #### __A4-3:__
>
> Please kindly refer to the response to all reviewers about limitations.
>
> #### __Reference:__
>
> [1] Dwibedi, Debidatta, et al. "With a little help from my friends: Nearest-neighbor contrastive learning of visual representations." arXiv preprint arXiv:2104.14548 (2021).
>
> [2] Baevski, Alexei, et al. "wav2vec 2.0: A framework for self-supervised learning of speech representations." arXiv preprint arXiv:2006.11477 (2020).
>
> [3] https://github.com/lightly-ai/lightly

---

> > ### Comment · Reviewer_mww5 · 2021-08-30
> > **Response**
> >
> > Thank you for the additional information. I will keep my score and recommend for acceptance.

---

> > > ### Author Response · Authors · 2021-08-31
> > > **Thank you for your response!**
> > >
> > > Thanks for recommendation of acceptance! We will include the additional information into the next version of our paper.

---

### Official Review · Reviewer_iJNi · 2021-07-14

**Rating:** 6
**Confidence:** 3

**Summary:**

The authors study the problem of contrastive learning, and in particular consider the NT-Xent loss function used in popular contrastive learning methods, such as the SimCLR algorithm. Such algorithms typically require long training times and very large batch sizes to be effective, and the authors explore this phenomenon in more detail.

Concretely, they identify a term in the gradient of this loss that has a potentially undesirable effect (the negative-positive coupling term), supported through empirical analysis. They adjust the contrastive loss function to prevent this term appearing in the resulting gradient, with the goal of improving learning speed and robustness to small batches.

In experiments on common SSL datasets including CIFAR10/100, STL10, and ImageNet, they find that by removing this coupling term, their method can improve significantly on competitive SimCLR and MoCo baselines, at much smaller batch sizes. Furthermore, they find that their method trains much faster with this adjustment to the loss.

Drawbacks: The authors do not appear to include a discussion of limitations in the paper or in the supplementary material. Furthermore, some of their baselines (in particular SimCLR) seems to be  weaker than other reported numbers in the literature, so this is worth examining in more detail.

**Limitations And Societal Impact:**

See above, but the authors do not mention limitations in their paper. This should be addressed.

**Main Review:**

# Originality
The paper contributes a nice mathematical argument as to what in the gradient of the  NT-Xent loss function might cause noisier learning (worse performance at small batch sizes and long training times). To the best of my knowledge, this is an original contribution, as is the specific adjustment to the loss function.

# Quality
The paper on the whole does a good job of identifying an important problem, hypothesising the cause (coupling term), providing some mathematical analysis and suggesting a solution, and then running empirical studies to verify their claims.

The experiments are fairly thorough, and consider important benchmark datasets used in the literature (both small and large scale).  Applying the method to both SimCLR and MoCo is a good idea, and it is  interesting to observe benefits in both settings. The improvement in learning speed is a very nice benefit of the method, and this was good to see analysed.

My main concern is that the baselines are weaker than numbers that I have seen reported in the literature. For example, consider this open source, popular implementation of SimCLR on CIFAR10: https://github.com/leftthomas/SimCLR. The reported result with kNN eval, batch size 512, and 500 epochs of training is 89.1%, which is significantly higher than numbers reported in the paper (best comparison: 81.3% in Table 1. This is 200 epochs, not 500, but I am not sure if we should expect an ~8% improvement with extra training alone).

Although the results are impressive, I would like to see the proposed method improve on a stronger SimCLR baseline, rather than one that seems to be fairly weak, to be more confident that the method truly is effective.

Relatedly, I am interested to understand why the performance of the proposed method appears to worsen with increased batch sizes in some situations. Is this just variation across seeds, or something more fundamental?

It would also help if the authors provided confidence intervals/measure of spread for the numbers reported. They mention running 5 repeats, and a measure of spread would be helpful (especially in understanding things like the above).

If computationally feasible, I am interested in the authors running an experiment with DCL with a large batch size (\~1024) and large number of training epochs (\~500) to understand if the performance degrades at these scales. It is good to understand if that might be a limitation of the method. I understand it is not a central focus of the paper, but it would be good to address this.

Furthermore, the authors mention they ran ablations in section 4.4. Can they provide more information about these? Was it just the reduction in learning epochs?

# Clarity
The paper is well written and laid out, and was very easy to read.

# Significance
Potentially high, but some outstanding questions regarding weak baselines, scalability to large batches and training times.

# Summary
Overall, I think this is a good contribution, but the authors do not discuss limitations of the work and the quality of the baselines is somewhat uncertain. If the authors can address these points, I will consider acceptance. For now, I think this is slightly below acceptance.

# Update post author response
Thank you for responding to my questions and providing the additional results. I am more encouraged by the method's performance now, and appreciate the discussion on limitations. It would be good for the authors to add all the relevant extra discussion into the final version of the paper (better comparisons to related work and additional experimental results). Given this and also the responses to the other reviewer comments, I am increasing my score accordingly.

**Time Spent Reviewing:**

4

---

> ### Author Response · Authors · 2021-08-10
> **Response to Reviewer iJNi**
>
>
> #### __Q3-1:__
>  My main concern is that the baselines are weaker than numbers that I have seen reported in the literature. For example, consider this open source, popular implementation of SimCLR on CIFAR10: https://github.com/leftthomas/SimCLR. The reported result with kNN eval, batch size 512, and 500 epochs of training is 89.1\%, which is significantly higher than numbers reported in the paper (best comparison: 81.3\% in Table 1. This is 200 epochs, not 500, but I am not sure if we should expect an ~8\% improvement with extra training alone).
>
> #### __A3-1:__
> Thanks for making an important point! This is a slight misunderstanding, and we will revise accordingly to avoid this confusion. For a fair comparison with CLD [1] on CIFAR10 data, we used the ResNet-18 as the backbone instead of the ResNet-50 in open source (https://github.com/leftthomas/SimCLR) or other literature (e.g., SimCLR paper). This ResNet-18 choice makes the SimCLR baselines look relatively weaker than those based on ResNet-50. While we have stated this on line-157, we realize that this could have been made clearer.
>
> To show the comparisons based on the ResNet-50 backbone, we adopt the GitHub implementation (https://github.com/leftthomas/SimCLR), using the same backbone and hyperparameters. The DCL model with kNN eval, batch size 512, and 500 epochs of training could reach 90.3\% compared to 89.1\%, which is reported in the GitHub repo.
>
> To address the reviewer’s concern more thoroughly, we show DCL ResNet-50 performance on CIFAR10 and CIFAR100 following the same experiment settings given in the open-source [2]. In these comparisons, we vary the batch size to show the effectiveness of DCL.
>
> CIFAR-10 comparisons of SimCLR baseline [2] and DCL with the ResNet-50, 500 training epochs, kNN eval.
>
> | Batch Size | 32 | 128 | 512 |
> | -------- | -------- | -------- | -------- |
> | SimCLR baseline      | 82.2\%     | 88.5\%     | 89.1\%     |
> | DCL      | 86.1\% (+3.9\%)   | 89.9\% (+1.4\%) | 90.3\% (+1.2\%) |
>
> CIFAR-100 comparisons of SimCLR baseline in [2] and DCL with the ResNet-50, 500 training epochs, kNN eval.
>
> | Batch Size | 32 | 128 | 512 |
> | -------- | -------- | -------- | -------- |
> | SimCLR baseline      | 49.8\%  | 59.9\% | 61.1\%    |
> | DCL      | 54.1\% (+4.3\%) | 61.6\% (+1.7\%) | 62.2\% (+1.1\%) |
>
> #### __Q3-2:__
> Although the results are impressive, I would like to see the proposed method improve on a stronger SimCLR baseline, rather than one that seems to be fairly weak, to be more confident that the method truly is effective.
>
> #### __A3-2:__
> As we have mentioned earlier, we believe this is a slight misunderstanding. Those ‘weak’ numbers were from a ResNet-18 backbone rather than a ResNet-50. Also based on this comment, we have slightly improved the DCL model performance on ImageNet-1K: 1) better hyperparameters, temperature $\tau = 0.2$ and learning rate $lr = 0.07$; 2) stronger augmentation (e.g., BYOL). We conduct an empirical hyperparameter search with batch size 256 and 200 epochs to obtain a stronger baseline. This improves DCL from 65.9\% to 67.8\% top-1 accuracy on ImageNet-1K. We further adopt the stronger augmentation policy from BYOL and improve our DCL from 67.8\% to 68.2\% top-1 accuracy on ImageNet-1K. This slightly improved result will be reported in the revision.
>
> | ImageNet-1K (batch size = 256; epoch = 200)   | Linear Top-1 Accuracy |
> | :-----------:| :-----------:     |
> | DCL         |   65.9\%   |
> | + optimal ($\tau$, $lr$) = (0.2, 0.07) | 67.8\%  (+1.9\%) |
> | + stronger augmentation | 68.2\% (+0.4\%) |
>
>
> #### __Q3-3:__
> Relatedly, I am interested to understand why the performance of the proposed method appears to worsen with increased batch sizes in some situations. Is this just variation across seeds, or something more fundamental?
>
> #### __A3-3:__
> We thank the reviewer for pointing this out. We hypothesize that when the batch size doubles, the training steps would reduce by a factor of two. This might be problematic on smaller datasets since there would not be many training steps overall in 500 epochs. We use the lr scaling method according to [3]; this might not be the optimal setting for small datasets like CIFAR-10 or STL-10. Notably, the SimCLR baseline also has this issue after batch size 1024. Since DCL achieves much better performance on small batch sizes, this issue starts to be more obvious at 512.
>
> #### __Q3-4:__
> It would also help if the authors provided confidence intervals/measure of spread for the numbers reported. They mention running 5 repeats, and a measure of spread would be helpful (especially in understanding things like the above).
>
> #### __A3-4:__
> This is a good suggestion! We will provide confidence intervals/measures of spread for the reported numbers in the revision for the experiments we can afford.
>
> #### __Q3-5:__
> If computationally feasible, I am interested in the authors running an experiment with DCL with a large batch size (\~1024) and large number of training epochs (\~500) to understand if the performance degrades at these scales. It is good to understand if that might be a limitation of the method. I understand it is not a central focus of the paper, but it would be good to address this.
>
> #### __A3-5:__
> We agree with the reviewer's view to address the results of DCL with large batch size and a large number of training epochs. According to the request from the reviewer, we provide the following results. The first two rows are taken from the appendix. It might be a limitation of the method when batch size is scaled up. We will address this carefully in the revision.
>
>
> | ImageNet-1K  | Epoch | Batch Size | Linear Top-1 Accuracy |
> | :-----------:| :-----------:  | :-----------: | :-----------: |
> | DCL          |   200   | 256  | 67.8\% |
> | DCL          |   400   | 256  | 69.5\%  |
> | DCL          |   400   | 1024 | 69.9\% |
>
>
> #### __Q3-6:__
> Furthermore, the authors mention they ran ablations in section 4.4. Can they provide more information about these? Was it just the reduction in learning epochs?
>
> #### __A3-6:__
> We thank the reviewer for the suggestion and this is our oversight. We will further provide extensive analysis on temperature $\tau$ in the loss to support that the DCL method is not sensitive to hyperparameters compared against the baselines. In the following, show the temperature $\tau$ search on both DCL and SimCLR baselines on CIFAR10 data. Specifically, we pretrain the network with temperature $\tau$ in {0.07, 0.1, 0.2, 0.3, 0.4, 0.5} and report results with kNN eval, batch size 512, and 500 epochs. As shown in the Table, compared to the SimCLR baseline, DCL is less sensitive to hyperparameters, e.g., temperature $\tau$.
>
> | Temperature __$\tau$__ 	|  0.07  	|   0.1  	|   0.2  	|   0.3  	|   0.4  	|   0.5  	| Standard deviation 	|
> |:----------------------:	|:------:	|:------:	|:------:	|:------:	|:------:	|:------:	|:------------------:	|
> |    Baseline (SimCLR)   	| 83.6\% 	| 87.5\% 	| 89.5\% 	| 89.2\% 	| 88.7\% 	| 89.1\% 	|       2.04\%       	|
> |           DCL          	| 88.3\% 	| 89.4\% 	| 90.8\% 	| 89.9\% 	| 89.6\% 	| 90.3\% 	|       __0.78\%__       	|
>
> If there is any other parameter search suggestion from the reviewer, we would appreciate it and revise accordingly.
>
>
> #### __Q3-7:__
> Limitations And Societal Impact:
> See above, but the authors do not mention limitations in their paper. This should be addressed.
>
> #### __A3-7:__
>
> Please kindly refer to the response to all reviewers about limitations.
>
> Reference:
>
> [1] Wang, Xudong, Ziwei Liu, and Stella X. Yu. "Unsupervised Feature Learning by Cross-Level Instance-Group Discrimination." Proceedings of the IEEE/CVF Conference on Computer Vision and Pattern Recognition. 2021.
>
> [2] https://github.com/leftthomas/SimCLR
>
>
> [3] Goyal, Priya, et al. ``Accurate, large minibatch sgd: Training imagenet in 1 hour." arXiv preprint arXiv:1706.02677 (2017).

---

> > ### Author Response · Authors · 2021-08-24
> > **We thank Reviewer iJNi for the update**
> >
> > Thank you for your suggestion and for reading our response. We will add both better comparisons to related work and additional experimental results into the next version of our paper.

---

### Official Review · Reviewer_K21n · 2021-07-14

**Rating:** 6
**Confidence:** 4

**Summary:**

This paper observes that in contrastive leanring the gradient contribution from positive pairs and negative pairs is scaled by a common multiplier that tends to be small in the small negative batch size N regime, and close to 1 for large N. The authors propose a minor modification to the InfoNCE loss function - removing the term on the denominator pertaining to the positive pair - that suffices to remove the multiplier (i.e. set it equal to 1 always). Although the change is minor, the authors observe an improvement in robustness to batch size in SimCLR (and queue size in MoCo) and find that fewer epochs of training are required to attain a certain level of performance (though in the large epoch regime the performance saturates at a similar optimal performance level as the vanilla InfoNCE loss).

**Limitations And Societal Impact:**

The authors do not discuss limitations or possible societal impact. Since the work is on fundamental methodology I am not hugely worried about the societal impact part. I would, however, appreciate a discussion of the limitations (especially since in the checklist they claim that they do include such a discussion in the appendix, for me only to find that there wasn't one).

The main limitation I would put forward is one discussed above - that even with the proposed cancellation in the loss, there still remains a fundamental tension between the alignment and uniformity terms in all contrastive losses, and that this may well be the main factor driving the large training time requirements of contrastive learning. Note: I am not trying to say that the authors have to resolve this limitation, but the purpose of the limitations discussion is to give the reader a more honest idea of where the future directions for further improvement may lie.

**Main Review:**


This paper has some merits. The motivation of reducing the coupling of gradient information from negative and positive samples is reasonable, the method sensible and very simple to implement, and the empirical results are encouraging.

However, despite those merits, I also have concerns that make me unwilling to recommend acceptance.  The objective derived is similar in form to the alignment and uniformity loss [1]. In [1] the order of the expectation and exponential is swapped, but the *key feature* of removing the positive term in the denominator is there. If you computed the gradients of the alignment and uniformity loss you would also find that it avoids the cancellation effect that this work posits as its main goal to remove.  It is *critically important* in my mind that [1] is discussed in the related work, and added as a baseline in experiments (especially for Fig. 3 on sensitivity to batch size and Fig. 4 on convergence speed). I will not feel confident enough to recommend acceptance without that comparison.

I strongly suspect that the objective of [1], if tuned correctly, would yield similar results to those in this paper (if I have missed some fundamental difference between the objective given here and that in [1] then please point it out - I will listen carefully and update my review on this point if needed). As a result of this, I would argue that the main contribution of this paper is not the proposed decoupled loss itself, but the observation that the loss can be motivated from the point of view of removing this gradient multiplier. While I think this is an interesting observation, on its own I do not feel it is sufficient contribution by the standards of NeurIPS.

---

## Minor criticisms:

- Somewhat slapdash reporting of results - e.g. Figure 4 doesn’t even tell the reader which dataset corresponds to which figure. Figure 1(a) talks about something called the “coefficient of variation” - I still don’t know what this is since I could not find an explanation anywhere in the text. Even if the explanation is hidden away somewhere, it shouldn't bee so hard for a reader to find it (do you just mean variance/standard dev.?). Please do a review of the pedagogical basics and make sure all figures are properly labeled and explained.
- In Figure 3 why use MoCo-v1 when you can just as easily run MoCo-v2 experiments and obtain more competitive baseline comparisons?
- Figure 3(b) has little practical consequence - in MoCo a larger memory bank is very cheap and generally no one feels a great need to shrink the size of the memory bank. Fig 3(a) on SimCLR is more interesting since batch size is a genuine constraint/consideration when using SimCLR.
- Table 2: baseline results are somewhat weak. SimCLR/MoCo-v2 results for CIFAR10 & STL10 results can, with a little care, be in the 90s (here they are generally in the low 80s) and CIFAR100 results in the high 60s or 70s (here they are in the 50s). This is not a major issue, and isn’t a deciding factor by any means, but stronger baselines are always preferable.
- The reweighed loss version L_DCW is not much of an innovation. Previous work [2,3] apply almost identical re-weighting schemes (to positive and negative pairs). There is no mention of this prior work in the text.

---

## Clarity:

This section does not form a component of my final assessment of this work as a research contribution, but is intended to help the authors with improving clarity.
- Clarity is fairly poor in general. Ideas that should be easy and intuitive are written in a convoluted way and end up harder than necessary to understand. I would especially suggest revising section 3, which contains the main idea of the paper.
- l.92 “Simsam” —> SimSiam
- Fig1 is on page 2, but relates to formulae & discussions on Page 4. I would suggest moving the figure to a more logical position.
- prop 2 is stated in a slightly odd (inverted) way. I would suggest a more linear sequence of logic: 1) define the new loss, 2) state that the new loss has gradients that are the same as the original loss, but with the “NPC multipliers” removed.

---

**Possible directions for improvement:**

(This component does not factor into the assessment, but attempts to give constructive suggestions on possible directions)

The main goal of this work - making contrastive learning more efficient in terms of both training time and memory requirements - is an interesting one that many researchers have thought about. While perhaps the multiplier may play a role in this, there still remains the fundamental issue that the “alignment” loss and the “uniformity” are trying to enforce *very different* goals. The large epoch requirement for contrastive learning is (in my opinion) likely due to the conflict between these two learning objectives. Its would be very interesting to see an extension of this work to not just discuss scalar multipliers in front of gradient terms, but also discuss conflicting gradient *directions* (the alignment terms gradient likely points in a conflicting direction from the uniformity terms gradient).

----

**Update 23rd Aug:**

Score moved from 5 to 6 based on information provided in the rebuttal.

---

**References:**

[1] Understanding contrastive representation learning through alignment and uniformity on the hypersphere, Wang and Isola

[2] Contrastive learning with hard negative samples, Robinson et al.

[3] Contrastive Attraction and Contrastive Repulsion for Representation Learning, Zheng et al.


**Time Spent Reviewing:**

3hrs

---

> ### Author Response · Authors · 2021-08-10
> **Response to Reviewer K21n**
>
> #### __Q2-1:__
> “However, despite those merits, I also have concerns .. The objective derived is similar .. to .. [1]. In [1] the order of the expectation and exponential is swapped, but the key feature of removing the positive term in the denominator is there. .. you would also find that it avoids the cancellation effect .. It is critically important in my mind that [1] is discussed in the related work, and added as a baseline in experiments .. ”
>
> #### __A2-1:__
> This is a very inspiring question, and thanks for your insightful comments! This leads us to a deeper understanding of the connection and difference between DCL and [1]. There is indeed a critical difference between DCL and [1], and the difference is exactly due to that the order of the expectation and exponential is swapped! Let’s assume the latent embedding vectors $z$ are normalized, for analytical convenience. When $z_i, z_j$ are normalized, $\exp\left(\langle z_i^{(k)}, z_i^{(l)}\rangle/\tau\right)$ and $\exp\left(-\lVert z_i^{(k)}- z_i^{(l)}\rVert^2/\tau\right)$ are the same, except for a trivial scale difference. Thus we can write $L_{DCL}$ and $L_{[1]}$ in a similar fashion:
> $$L_{DCL} = L_{DCL,pos} + L_{DCL, neg}, L_{DCL, neg} = \sum_i{\log\left(\sum_{j\neq i}{\exp\left(\langle z_i^{(k)}, z_i^{(l)}\rangle/\tau\right)}\right)}$$
> $$L_{[1]} = L_{align} + L_{uniform},\ L_{uniform} = \log\left(\sum_i \sum_{j\neq i}{\exp\left(\langle z_i^{(k)}, z_i^{(l)}\rangle/\tau\right)}\right) $$
>
> With the right weight factor, $L_{align}$ can be made exactly the same as $L_{DCL,pos}$. So let’s focus on $L_{DCL, neg}$ and $L_{uniform}$:
> $$L_{DCL, neg} = \sum_i{\log\left(\sum_{j\neq i}{\exp\left(\langle z_i^{(k)}, z_i^{(l)}\rangle/\tau\right)}\right)}$$
> $$L_{uniform} = \log\left(\sum_i \sum_{j\neq i}{\exp\left(\langle z_i^{(k)}, z_i^{(l)}\rangle/\tau\right)}\right)$$
>
> Similar to our analysis in the manuscript, the latter $L_{uniform}$ introduces a negative-negative coupling between the negative samples of different positive samples. If two negative samples of $z_i$ are close to each other, the gradient for $z_i$ would also be attenuated. This behaves similarly to the negative-positive coupling. That being said, while [1] doesn’t have a negative-positive coupling, it has a similarly problematic negative-negative coupling. We will revise our manuscript to reflect this analysis to compare to [1]. Further, we will provide a comprehensive empirical comparison to show DCL outperforms [1]. The empirical experiments match our analytical prediction: DCL outperforms [1] with a larger margin under a smaller batch size.
>
>
> #### __Q2-2:__
> I strongly suspect that the objective of [1], if tuned correctly, would yield similar results to those in this paper ...
>
> #### __A2-2:__
> In this experiment, we compare DCL to [1] on STL-10, ImageNet-100, ImageNet-1K under various settings.
>
> For STL-10 data, we implement DCL based on the official code of [1] (https://github.com/SsnL/align_uniform/tree/master/examples/stl10). The backbone encoder and the hyperparameters are the same as the implementation of [1], which has not been optimized for DCL in any way. The reviewer probably has found that [1] has done a quite thorough hyperparameter search. We do not even think we can easily surpass the parameter tuning in [1]. :p Please kindly refer to Table -8 to -11 in [1]. We believe the default hyperparameters are relatively optimized for [1].
>
> DCL reaches 84.4\% (fc7+Linear) compared to 83.2\% (fc7+Linear) reported in [1]. Again, please note that we did not tune the parameters for DCL at all. This should be a more than fair comparison.
>
> STL-10 comparisons of [1] and DCL under the same experiment setting.
>
> | | fc7 + Linear| fc7 + 5-NN | Output + Linear | Output + 5-NN |
> | :-----------: | :-----------: | :-----------: | :-----------: | :-----------: |
> | [1]     |   83.2\%     | 76.2\%     | 80.1\% | 79.2\%    |
> | DCL     |   84.4\% (+1.2\%) |77.3\% (+1.1\%) |81.5\% (+1.4\%) |80.5\% (+1.3\%)
>
>
> ImageNet-100 comparisons of [1] and DCL under the same setting (MoCo).
>
> | | Epoch | Memory Queue Size | Linear Top-1 Accuracy |
> | :-----------:| :-----------: | :-----------: | :-----------: |
> | [1]          |   240   | 16384   | 75.6\% |
> | DCL          |   240   | 16384   | 76.8\% (+1.2\%) |
>
>
> ImageNet-1K comparisons of [1] and DCL under the best setting. In this experiment both of the methods used their optimized hyperparameters.
>
> |  | Epoch   | Batch Size | Linear Top-1 Accuracy |
> | :-----------:| :-----------:   | :-----------: | :-----------: |
> | [1]          |   200   | 256 (Memory queue = 16384)   | 67.69\% |
> | DCL          |   200   | 256   | 68.2\% (+0.51\%) |
>
> STL-10 comparisons of [1] and DCL under different batch sizes.
>
> |Batch Size       | 32   | 64 | 128 | 256 | 768 |
> | :-----------:| :-----------:   | :-----------: | :-----------: | :-----------: | :-----------: |
> | [1]         | 78.9\%     | 81.0\%   | 81.9\% | 82.6\%| 83.2\%|
> | DCL        |  81.0\% (+2.1\%) |  82.9\% (+1.9\%) | 83.7\% (+1.8\%) | 84.2\% (+1.6\%) | 84.4\% (+1.2\%) |
>
> In every single one of the experiments, DCL outperforms [1]. The last experiment shows the advantage of DCL becoming larger with a smaller batch size. We hope these results show the unique value of DCL compared to [1].
>
> #### __Q2-3: Minor criticisms__
>
> __Q:__ Somewhat slapdash reporting of results …
>
> __A:__ Thank you for pointing this out! We will revise the figure caption accordingly.
>
> __Q:__ “Figure 1(a) talks about something called the ``coefficient of variation’’ ...
>
> __A:__  The coefficient of variation, $c_v$ is a standardized measure of the dispersion of a probability distribution. We will provide the definition in the text and legend of the figure, which is the ratio of the standard deviation $\sigma$ to the mean $\mu$, $c_v = \frac{\sigma}{\mu}$.
>
> __Q:__ In Figure 3 why use MoCo-v1 when you can just as easily run MoCo-v2 experiments and obtain more competitive baseline comparisons?
>
> __A:__ Based on your suggestion, we implement DCL on MoCo-V2. It is worth mentioning that the DCL achieves competitive performance with more efficiency (smaller memory queue). We will provide more comprehensive results in the revision.
>
> ImageNet-100 comparisons of [1] and DCL under the same setting (MoCoV2) except for memory queue size.
>
> | ImageNet-100 | Epoch | Memory Queue Size | Linear Top-1 Accuracy |
> | :-----------:| :-----------: | :-----------: | :-----------: |
> | [1]            |   200   | 16384   | 77.66\% |
> | DCL         |   200   | 8192   | 80.52\% (+2.86\%) |
>
>  __Q:__ Figure 3(b) has little practical consequence ...
>
> __A:__ We agree with this point on the technical side. In our view, we focus on the theoretic analysis instead of just the technical problem. DCL is implemented on different contrastive-based methods to show its generalization. Although the memory bank in MoCo is cheap, our main goal is to verify that the theoretical insight could effectively improve the performance of contrastive-based approaches with fewer negative samples. A model with less assumptions and equal performance is better.
>
>  __Q:__ Table 2: baseline results are somewhat weak. ...
>
> __A:__ Thanks for making an important point! This is a slight misunderstanding, and we will revise accordingly to avoid this confusion. For a fair comparison with CLD [1] on CIFAR data, we used the ResNet-18 as the backbone instead of the ResNet-50 in other literature (e.g., SimCLR paper).
> To address the reviewer’s concern more thoroughly, we show DCL ResNet-50 performance on CIFAR10 and CIFAR100 following [2]. In these comparisons, we vary the batch size to show the effectiveness of DCL.
>
> CIFAR-10 comparisons of SimCLR baseline and DCL, kNN eval.
>
> | Batch Size | 32 | 128 | 512 |
> | -------- | -------- | -------- | -------- |
> | SimCLR baseline      | 82.2\%     | 88.5\%     | 89.1\%     |
> | DCL      | 86.1\% (+3.9\%)   | 89.9\% (+1.4\%) | 90.3\% (+1.2\%) |
>
> CIFAR-100 comparisons of SimCLR baseline and DCL, kNN eval.
>
> | Batch Size | 32 | 128 | 512 |
> | -------- | -------- | -------- | -------- |
> | SimCLR baseline      | 49.8\%  | 59.9\% | 61.1\%    |
> | DCL      | 54.1\% (+4.3\%) | 61.6\% (+1.7\%) | 62.2\% (+1.1\%) |
>
> __Q:__ Previous work [2,3] apply almost identical re-weighting schemes ...
>
> __A:__ We agree with the point of the reviewer that the reweighted version of DCL might not have any strong innovation. The main focus of our method is on the positive and negative decoupling. We will have a detailed discussion of previous work [2, 3] in the revised version.
>
> #### __Q2-4: Clarity__
> __A:__ Thank you for pointing out the typos and the suggestions! We will improve the clarity in the revision accordingly.
>
> #### __Q2-5: Possible directions for improvement__
> . .. Its would be very interesting to see an extension of this work to not just discuss scalar multipliers in front of gradient terms, but also discuss conflicting gradient directions ..”
>
> #### __A2-5:__
> We greatly appreciate these suggestions! We implemented the DCL with a large epoch, and the results are in the Table below. Also, as contrastive methods, the conflicting gradient is indeed an interesting future research direction, though self-supervised learning for visual analysis remains under-explored in general.
>
> | ImageNet-1K  | Epoch | Batch Size | Linear Top-1 Accuracy |
> | :-----------:| :-----------:  | :-----------: | :-----------: |
> | DCL          |   200   | 256   | 67.8\% |
> | DCL          |   400   | 256   | 69.5\% |
> | DCL          |   400   | 1024 | 69.9\%|
>
> #### __References:__
>
>  [1] Wang and Isola, Understanding contrastive representation learning through alignment and uniformity on the hypersphere
>
>  [2] Robinson et al. Contrastive learning with hard negative samples
>
>  [3] Zheng et al. Contrastive Attraction and Contrastive Repulsion for Representation Learning
>
>  [4] Wang et al.  "Unsupervised Feature Learning by Cross-Level Instance-Group Discrimination." CVPR 2021.
>
>  [5] https://github.com/leftthomas/SimCLR

---

> > ### Comment · Reviewer_K21n · 2021-08-23
> > **Thanks for the additional information**
> >
> > Dear authors,
> >
> > Thanks for your efforts in responding to the points raised in my review. I am particularly grateful for the added comparison between your method and [1]. Because this was my main concern, and your rebuttal has, in my view, adequately addressed this comparison, I am happy to move my review up a point to 6. This update is of course on the proviso that the results and mathematical discussion of the distinction between your method and [1] is included in the updated paper.
> >
> > Best wishes

---

> > > ### Author Response · Authors · 2021-08-24
> > > **Thank you for your review!**
> > >
> > > Thank you for reading our response and recognizing our contributions. We will incorporate both the results and mathematical discussion between our method and [1] in the next version of our paper.

---

### Official Review · Reviewer_taNp · 2021-07-16

**Rating:** 6
**Confidence:** 4

**Summary:**

The authors propose an improved loss based on the infoNCE loss and show that SimCLR and MoCo with the proposed loss converge faster and require smaller batch sizes / memory banks.

**Limitations And Societal Impact:**

The authors claim the limitations to be addressed in supplementary material, but I didn't find any. The potential negative societal impact is not discussed.

**Main Review:**

My main concern about the paper is that given there are so many papers discussing the losses for contrastive learning, the authors did not cite any of them [1-3]. Some of these papers also focus on batch size sensitivity. I would recommend authors do a more comprehensive literature search and compare the proposed loss with these previous losses.

The thermotical analysis provides interesting insights but the authors seem not to claim it in the right way. Based on Figure 1 and my understanding, small batch sizes lead to a smaller chance to utilize hard negatives in each iteration, and consequently a simpler problem to optimize, i.e. $q_B \rightarrow 0$. Removing $q_B$ simply increases the signals provided by the simple negatives and positives. Instead, the authors claim the main issue to be a negative-positive coupling problem, which is not clear to me why that impacts the training.

The authors do not provide the code. Although it is simple to reproduce and the authors mention that this loss is insensitive to the hyperparameters, they do not provide any evidence to support this other than the claims in 4.4.

The authors state in the checklist that "they describe the limitations of the work in supplementary material", but they don't.

The empirical results of this paper are strong and the theoretical analysis is interesting. However, the paper lacks several relevant references. And there is evident space for the authors to improve in both experiments and writing. Therefore, I'm inclined to reject the paper at the current stage.

Minor issues:

1. line 14: Moco --> MoCo

2. Figure 1: $q_b$ --> $q_B$

3. line 211: rely --> relies

4. line 271-272: please delete the official template.


[1] Tsai, Yao-Hung Hubert, et al. "Self-supervised Representation Learning with Relative Predictive Coding." ICLR 2021.
[2] Hjelm, R. Devon, et al. "Learning deep representations by mutual information estimation and maximization." ICLR 2019.
[3] Ozair, Sherjil, et al. "Wasserstein Dependency Measure for Representation Learning." NeurIPS 2019.


**Time Spent Reviewing:**

7

---

> ### Author Response · Authors · 2021-08-10
> **Response to Reviewer taNp**
>
> #### __Q1-1:__
> My main concern ..is that given there are so many papers discussing the losses for contrastive learning, the authors did not cite any of them [1-3]. Some of these papers also focus on batch size sensitivity. I would recommend authors do a more comprehensive literature search and compare the proposed loss with these previous losses.
>
> #### __A1-1:__
> We thank the reviewer for pointing out these valuable references! Our starting point comes from SimCLR and then provides theoretical analysis to support why decoupling the positive and negative terms in contrastive loss (1) is essential. [1] starts from the contrastive predictive code $J_{CPC}$, which is equivalent to equation (1), and then proposes a new term $J_{RPC}$. However, $J_{RPC}$ is not the same as (1) or $J_{CPC}$ in essence. $J_{RPC}$ is more similar to the ranking loss, which collects and pushes away the positive pairs and negative pairs. Since the ranking loss is not stable enough, [1] adds additional regularization terms to control the magnitude of the network and gains better results. On the other hand, it brings additional hyperparameters and needs more time to search for the best weight combinations. [2] follows the Mutual Information Neural Estimation (MINE) approach and extends the idea between the local and global features. Hence, [2] is quite different from the contrastive loss. [3] follows the approach of $J_{CPC}$ and proposes a Wasserstein distance to prevent the encoder from learning any other differences between unpaired samples. Although the motivations are similar, the target problems are different. Overall, we will add a discussion for [1-3] in the related work, emphasizing [1].
>
> #### __Q1-2:__
> The thermotical analysis provides interesting insights but the authors seem not to claim it in the right way. Based on Figure 1 and my understanding, small batch sizes lead to a smaller chance to utilize hard negatives in each iteration, and consequently a simpler problem to optimize, i.e. $q_B \rightarrow 0$ . Removing $q_B$ simply increases the signals provided by the simple negatives and positives. Instead, the authors claim the main issue to be a negative-positive coupling problem, which is not clear to me why that impacts the training.
> #### __A1-2:__
> We would agree with the reviewer’s reasoning. And, we see no conflict between our reasoning and the reasoning from the reviewer. Rather, the reviewer clarifies the same insight as us but with another interpretation from a different perspective, which helps us build a deeper understanding. When the negatives are easy, the coupling multiplier will reduce the gradient. Vice versa, when the positive sample is easy, the coupling multiplier would also reduce the gradient. So essentially, when the classification task is easy in SimCLR, the gradient would be attenuated by the NPC multiplier. However, the easiness is relative in SimCLR and MoCo. Given a hard negative, an easier positive would reduce the learning signal as well. The DCL objective removes this NPC multiplier so that the mutual impact between the positive and negative samples is removed. Thus, our reasoning aligns. We will extend the discussion accordingly.
>
> #### __Q1-3:__
> The authors do not provide the code.
>
>
> #### __A1-3:__
> The method is indeed straightforward to reproduce, as it has been confirmed by several of our colleagues. For anyone who works with SimCLR or MoCo, the essential thing to do is remove a single term, which requires a few lines of code. Since the method is not sensitive to hyperparameters, it’s unnecessary to tune the parameters to see an initial performance gain. Given this reason, we did not feel any urgency to provide the code. In fact, if one already has a working SimCLR code, it shall be more efficient to modify it accordingly rather than starting from our implementation as it requires a few steps to set up the package environment.
>
> With that being said, we promise to release the code, which can easily reproduce several important experiments in the manuscript. Our implementation is partially based on the OpenSelfSup [4] framework, as we mentioned in the manuscript. Based on the feedback from our colleagues, setting up the environment might take longer than implementing the DCL loss on an existing SimCLR model. So we are migrating the implementation to an easier to set up version.
>
> #### __Q1-4:__
> Although it is simple to reproduce and the authors mention that this loss is insensitive to the hyperparameters, they do not provide any evidence to support this other than the claims in 4.4.
>
> #### __A1-4:__
> We thank the reviewer for the suggestion and this is our oversight. We will further provide extensive analysis on temperature $\tau$ in the loss to support that the DCL method is not sensitive to hyperparameters compared against the baselines. In the following, show the temperature $\tau$ search on both DCL and SimCLR baselines on CIFAR10 data. Specifically, we pretrain the network with temperature $\tau$ in {0.07, 0.1, 0.2, 0.3, 0.4, 0.5} and report results with kNN eval, batch size 512, and 500 epochs. As shown in the Table, compared to the SimCLR baseline, DCL is less sensitive to hyperparameters, e.g., temperature $\tau$.
>
> | Temperature __$\tau$__ 	|  0.07  	|   0.1  	|   0.2  	|   0.3  	|   0.4  	|   0.5  	| Standard deviation 	|
> |:----------------------:	|:------:	|:------:	|:------:	|:------:	|:------:	|:------:	|:------------------:	|
> |    Baseline (SimCLR)   	| 83.6\% 	| 87.5\% 	| 89.5\% 	| 89.2\% 	| 88.7\% 	| 89.1\% 	|       2.04\%       	|
> |           DCL          	| 88.3\% 	| 89.4\% 	| 90.8\% 	| 89.9\% 	| 89.6\% 	| 90.3\% 	|       __0.78\%__       	|
>
> If there is any other parameter search suggestion from the reviewer, we would appreciate it and revise accordingly.
>
> #### __Q1-5:__
> The authors state in the checklist that "they describe the limitations of the work in supplementary material", but they don't.
> #### __A1-5:__
>
> Please kindly refer to the response to all reviewers about limitations.
>
> #### __Q1-6:__
> The empirical results of this paper are strong and the theoretical analysis is interesting. However, the paper lacks several relevant references. And there is evident space for the authors to improve in both experiments and writing. Therefore, I'm inclined to reject the paper at the current stage.
> #### __A1-6:__
> We take the reviewers' comments into full consideration. First, we will discuss the suggested references accordingly in the revision. We further improve our DCL to 68.2\% top-1 accuracy on ImageNet-1K over 200 epochs and 69.5\% top-1 accuracy over 400 epochs. We will also improve the writing and report the slightly improved result as well.
>
> #### __Q1-7:__
> Minor issues:
>
> 1. line 14: Moco --> MoCo
>
> 2. Figure 1: $q_b$ --> $q_B$
>
> 3. line 211: rely --> relies
>
> 4. line 271-272: please delete the official template.
>
> #### __A1-7:__
>    Thanks for the detailed errata, and these will be fixed.
>
> #### __Q1-8:__
> Limitations And Societal Impact:
> The authors claim the limitations to be addressed in supplementary material, but I didn't find any. The potential negative societal impact is not discussed.
>
> #### __A1-8:__
>
> Please kindly refer to the response to all reviewers about limitations.
>
> ####  __Reference:__
>
>  [1] Tsai, Yao-Hung Hubert, et al. "Self-supervised Representation Learning with Relative Predictive Coding." ICLR 2021.
>
>  [2] Hjelm, R. Devon, et al. "Learning deep representations by mutual information estimation and maximization." ICLR 2019.
>
>  [3] Ozair, Sherjil, et al. "Wasserstein Dependency Measure for Representation Learning." NeurIPS 2019.
>
>  [4] https://github.com/open-mmlab/OpenSelfSup

---

> > ### Comment · Reviewer_taNp · 2021-08-27
> > **The response addresses my concern well and therefore I have raised my score**
> >
> > I would like to thank the authors for their comparison with the previously proposed losses and additional experiments. My concern on the reproductivity is solved based on the new experiment results. Therefore, I would like to raise my score to 6 to reflect this.

---

> > > ### Author Response · Authors · 2021-08-28
> > > **Thank you for your review!**
> > >
> > > Thank you for your comments and for reading our response. We will add the expanded results to the next version of our paper.

---

### Author Response · Authors · 2021-08-10
**Response to All Reviewers about Limitations**

We thank all reviewers for kind reminder of the limitations. We want to summarize two main limitations of the proposed DCL method. First, the performance of DCL appears to have a relatively minor gain compared to the baseline when the batch size is large. According to Figure 1 in the manuscript and the theoretical analysis,  the reason is that the NPC multiplier $q_B \rightarrow 1$ while the batch size is increased to a relatively huge number (e.g., 1024). So the baseline SimCLR loss converges to the DCL loss as the batch size approaches infinity. Second, the scenario of DCL focuses on contrastive learning-based methods, where we decouple the positive and negative terms to achieve better performance. However, non-negative methods, e.g., BYOL, do not rely on negative samples. We will make these limitations clearer in the revision.

---

### Decision · Program_Chairs · 2021-09-27

**Decision:**

Reject

**Comment:**

This paper presents an analysis of contrastive learning with the InfoNCE objective that identifies a strong dependence of the gradient on batch size, and proposes a new objective to remove this batch-size dependence by eliminating the positive pair from the denominator of the InfoNCE loss. This decouples the influence of the positive and negative pairs on the loss function (L_DC) in an additive manner, which is not true for the original InfoNCE loss. Through extensive experiments (in the original paper as well as rebuttal), the effectiveness over InfoNCE at small batch size is verified.

Reveiewers raised several questions around the experimental setting and comparison to additional approaches (Barlow Twins, SimSiam, BYOL) that were largely addressed in the rebuttal. However, there remained concerns around the impact of yet another contrastive loss function given the success of non-contrastive losses like BYOL at small batch sizes. While the theoretical analysis and results in Fig 1 highlight the batch-size dependence of the gradient scale, there is not discussion of why removing this batch-size dependent impact should yield improved representation learning abilities, or how the new objective may be related to other proposed losses (but some of these concerns have been nicely addressed in the rebuttal, e.g. relation to alignment + uniformity losses).

[[Addressing these concerns and incorporating more results from the rebuttal into the main paper would greatly improve the paper, but I cannot recommend it for acceptance in the current state.]]